# Changes in Plant Diversity and Soil Factors under Different Rocky Desertification Degrees in Northern Guangdong, China

Mingyu Lan [1], Chunquan Xue [2], Jiazhi Yang [2], Ning Wang [1], Chuanxi Sun [1], Guozhang Wu [1], Hongyu Chen [1] and Zhiyao Su [1,*]

1   College of Forestry and Landscape Architecture, South China Agricultural University, Guangzhou 510642, China
2   Guangdong Forestry Survey and Planning Institute, Guangzhou 510520, China
*   Correspondence: zysu@scau.edu.cn; Tel.: +86-20-8528-0263

**Abstract:** Revegetation is an important restoration strategy for the control of rocky desertification. However, few studies have focused on the effects of different rocky desertification degrees (RDDs) on plant diversity and soil fertility in northern Guangdong over long periods of time. In this study, variance analysis, correlation analysis, and canonical correlation analysis (CCA) were used to examine plant diversity, soil physicochemical properties, and their correlations in various rocky desertification areas in northern Guangdong. The results showed that the Pinaceae, Lauraceae, and Fagaceae species were relatively abundant in the rocky desertification areas of northern Guangdong. Additionally, *Cinnamomum camphora*, *Schima superba*, *Pinus massoniana*, *Quercus stewardiana*, and *Acer camphora* could be used as indicators for rocky desertification. There were significant differences in plant community compositions and diversity characteristics between the five RDDs, and the vegetation exhibited the trend of initial destruction and then gradual improvement and stabilization. There were significant differences in soil bulk density, mechanical composition, organic matter, total nitrogen, alkaline hydrolysis nitrogen, and available potassium between the different RDDs. Except for pH, the soil chemical characteristics all had clear aggregation effects. Soil organic matter, total nitrogen, total potassium, and alkaline hydrolysis nitrogen all exhibited degradation–improvement cycles. The correlation analysis revealed that there was a significant correlation between soil physicochemical properties and species diversity. The CCA analysis showed that the most important soil factors affecting plant community structures were total phosphorus and available phosphorus. In conclusion, some achievements have been made in the restoration of rocky desertification in northern Guangdong; while the plant community structure improved, some soil nutrients also improved. Vegetation and soil have a strong coupling relationship. In the later stages of recovery, suitable species for rocky desertification could be considered in varying degrees and P and K could be supplemented appropriately. Our study will have implications for the revegetation of rocky desertification.

**Keywords:** correlation; northern Guangdong; plant diversity; rock desertification; soil physicochemical properties

## 1. Introduction

Rocky desertification in China is currently one of the three most serious ecological issues posing threats to the region's ecology, economy, and environment [1]. Rocky desertification is a form of land degradation in subtropical karst environments that have been disturbed and destroyed by irrational socioeconomic human activity, leading to severe soil erosion, the exposure of a significant amount of bedrock, a significant decline in land productivity, and the emergence of desert-like landscapes [2,3]. The rocky desertification process upsets the soil–vegetation balance, squanders soil nutrients, impedes plant growth, and degrades the ecosystem [4]. China's rocky desertification areas are mainly distributed in the Hubei, Hunan, Guangdong, Guangxi, Chongqing, Sichuan, Guizhou,

and Yunnan provinces (autonomous regions and municipalities), within the scope of about 452,000 square kilometers [5]. With a total area of 81,329.8 hectares, the rocky desertification area in Guangdong province is primarily dispersed in the northern portion of the province, including Yingde, Lianzhou, Qingxin, Liannan, Yangshan, and other towns and counties [6]. The extremely fragile environmental conditions and the limitations of social and economic development make it difficult for the region to pursue sustainable development [7]. Thus, the control of rocky desertification is a pressing issue in China [8].

Plant diversity can reflect plant community structures, development stage, and stability [9,10]; therefore, studying plant diversity and its affecting factors in rocky desertification areas is crucial. The study of the diversity of rocky desertification plants has achieved a number of advances. Plant diversity and community compositions in different rocky desertification areas exhibit a variety of traits, but overall, the community structures tend to be straightforward, and the diversity index tends to be low [11,12]. According to previous research, artificially planted vegetation produces a higher species variety than other restoration techniques and its restoration impact is superior to that of air-seeded vegetation [13]. In artificial afforestation, mixed forest management strategies have better impacts on plant diversity restoration [14]. The study of plant diversity in various habitats within rocky desertification areas has discovered that vegetation is the most diverse in soil surface microhabitats, followed by stone ditches, and the least diverse in stone caves [15]. There have also been numerous studies on plant diversity in different rocky desertification degrees, but the outcomes have varied greatly depending on the region. The diversity index in the rocky desertification area in southwestern Hunan has exhibited a tendency to first decrease and then increase with the increase in rocky desertification, and species richness has shown the trend of steadily increasing [16]. However, as rocky desertification has progressed in Guizhou province, the evenness and dominance indices have increased, while the Shannon–Wiener index and species richness have decreased [11]. This demonstrates that there is regional heterogeneity in the changes in plant diversity among the different rocky desertification degrees; thus, measures to manage this process should be tailored to local conditions.

An essential part of terrestrial ecosystems is soil. In addition to influencing the structural aspects of plant communities, soil physicochemical properties also impact the regeneration and succession of those communities [17]. The characteristics of rocky desertification soil are thin soil cover and fast nutrient loss. Therefore, studying changes in soil factors during the process of rocky desertification could lay the foundations for artificially controlling plant community succession [18]. The soil conditions in rocky desertification areas have been found to be poor [19] and there are substantial differences in soil pH, SOM, TN, TP, and TK, depending on the rocky desertification degree. These indices initially decline as the rocky desertification degree increases, and then slowly improve [7]. Although the soil in rocky desertification areas has been studied extensively, the interactions between soil and other variables are also of crucial concern. The relationship between soil and plants is an indivisible whole. The former provides nutrients, water, and suitable temperature for plant growth, while the latter also helps the former to some extent [20]. It has been found that plant species richness [21] and soil nutrient contents [22] in rocky desertification areas are significantly lower than those in other areas and that there are significant correlations between soil factors and plant diversity [23,24]. Therefore, research into the interaction between vegetation and soil in rocky desertification areas could help to ameliorate the conditions of both the vegetation and soil, which could in turn help to control rocky desertification. So far, there have been numerous studies on the vegetation and soil in rocky desertification areas in southwestern China [11,25], but most of them have mainly concentrated on single changes in soil physicochemical properties [26] or plant diversity [27] during the rocky desertification process, and there have been few studies on the correlations between soil environments and plant communities.

In addition, environmental factors play a vital role in the development of rocky desertification and vegetation reconstruction [28]. At present, there are many studies on the

influencing factors of vegetation reconstruction and soil restoration in rocky desertification. As an important topographic factor, altitude is considered one of the decisive factors affecting the distribution pattern of species diversity [29,30]. Its change can change the hydrothermal conditions of the environment and increase the intensity of light, thus affecting the species composition and structure of plant communities. Some studies have found that high-altitude areas have sufficient light and intensified hydraulic erosion, which provide favorable conditions for the growth of plants suitable for rocky desertification [31]. Moreover, the human disturbance in high-altitude areas is relatively small, the relatively primitive ecological environment is maintained, and the degree of rocky desertification is relatively low [32]. Rainfall patterns can affect the characteristics and degree of soil erosion in karst rocky desertification [33], and soil erosion is one of the factors that cause the degradation of karst rocky desertification land [34]. In recent years, changes in precipitation patterns caused by global temperature changes have increased extreme precipitation events in karst rocky desertification areas, thus deepening soil erosion [35]. Studies have found that summer is the season with the highest precipitation and frequency in most regions, and therefore has the most obvious impact on soil erosion [36].

Scholarly interest in vegetation and soil restoration in rocky desertification areas has recently increased [37]. However, the majority of research has focused on the rocky desertification areas in southwestern China [38], while there have been relatively few studies on the rocky desertification areas in northern Guangdong. Additionally, most studies have mainly focused on the causes [32] and governance [39] of rocky desertification. The development of technical methods for plant restoration in these areas has been hampered by a lack of systematic research on vegetation changes and soil conditions during the process of rocky desertification. Therefore, in this study, we aimed to describe the plant diversity and soil factor variation patterns of different rocky desertification degrees. The objectives were to explore the following: (1) variation characteristics of plant diversity in different rocky desertification degrees, (2) variation characteristics of soil factors in different degrees of rocky desertification, and (3) relationship between plant diversity and soil factors in rocky desertification mountainous areas. This could systematically comprehend the ecosystem changes brought about by the rocky desertification process in northern Guangdong and it provides an important basis for revegetation in rocky desertification, which would be crucial for the prevention of rocky desertification.

## 2. Materials and Methods

### 2.1. Study Sites

The study areas were located in four rocky desertification areas in northern Guangdong: Lechang (113°02′ E–113°05′ E, 25°08′ N–25°12′ N), Yingde (112°47′ E–113°24′ E, 24°09′ N–24°22′ N), Ruyuan (113°07′ E–113°10′ E, 25°00′ N–25°06′ N), and Yangshan (112°36′ E–112°43′ E, 24°28′ N–24°36′ N). The region has a mid-subtropical–southern subtropical monsoon climate, with an average annual temperature of 15.5–22 °C and an average annual rainfall of 1500–2590 mm. The main types of landforms are flowing landforms and karst landforms. The karst landforms are mainly distributed in the northwest of Lechang City, northern Yingde, northeastern Ruyuan, and most of Yangshan. There are many kinds of vegetation in these areas [40].

### 2.2. Sample Plot Arrangement

According to Xiong Kangning's classification standard of RDD [41], the rocky desertification areas in northern Guangdong were divided into five degrees: potential (I), mild (II), moderate (III), severe (IV), and extremely severe (V). Then, 7 representative plots (20 × 20 m) were selected for each degree, producing a total of 35 plots. The basic information of the sample plots is shown in Table 1.

**Table 1.** The basic information of the sample plots. Note: LC, Lechang; YD, Yingde; RY, Ruyuan; YS, Yangshan.

| RDD | Plot Location | Bare Rock Ratio (%) | Soil Type | Average Altitude (m) | Average Slope (%) | Major Vegetation |
|---|---|---|---|---|---|---|
| I | LC, YD, RY, YS | <30 | Sandy loam | 322.3 | 11.5 | *Cinnamomum burmannii, Cinnamomum camphora, Pinus massoniana, Castanopsis sclerophylla, Osmanthus fragrans* |
| II | YD, YS | 30–50 | Loam | 260.0 | 18.2 | *Pinus massoniana, Cinnamomum burmannii* |
| III | LC, YD | 51–70 | Sandy loam | 508.5 | 16.4 | *Castanopsis sclerophylla, Machilus chinensis* |
| IV | YD, RY, YS | 71–90 | Loam | 547.9 | 21.4 | *Cornus wilsoniana, Celtis sinensis* |
| V | RY, YS | >90 | Loam | 472.0 | 28.6 | *Quercus stewardiana, Acer coriaceifolium* |

*2.3. Sample Plot Surveys and Analyses*

The typical community survey plots were set up in the study areas and the vegetation surveys were carried out in the first-level quadrats (20 × 20 m). All woody plants with a DBH greater than 1 cm at 1.3 m in the quadrat were investigated and their species name, DBH, tree height (H), and crown width (P) were recorded.

The importance value (IV) and species diversity indices of the woody layers were also calculated based on the field survey data. The species diversity indices included the species richness (R), Shannon–Wiener index (H), Pielou's index (J), and Berger–Parker index (D). The calculation methods were as follows:

$$IV_i = (RD_i + RF_i + RC_i)/3 \tag{1}$$

$$H' = -\sum_{i=1}^{S}(P_i \times InP_i) \tag{2}$$

$$J = H'/InS \tag{3}$$

$$D = N_{max}/N \tag{4}$$

where $RD_i$ is the relative density, that is, the ratio of the number of individuals of a species to the number of individuals of all species. $RF_i$ is the relative frequency, that is, the ratio of the frequency of a species to the sum of the frequency of all species. $RC_i$ is the relative advantage, that is, the ratio of the basal area of a species to the basal area of all species. N is the total number of individuals per species, S is the number of species, $P_i$ is the proportion of all individuals of species I in the total number of individuals, and $N_{max}$ is the number of individuals of the dominant species.

*2.4. Soil Sample Collection and Analysis*

According to the industry standard 'LY/T 1210-1999' [42] and 'LY/T 1215-1999' [43], five sampling points were selected in each 20 × 20 m quadrat using the five-point sampling method, and then soil samples (0–30 cm) were collected using a ring knife. The samples were placed into sealed bags, taken back to the laboratory, and dried in an oven at 105 °C to calculate the soil moisture content (SMC) and soil bulk density (SBD). The soil samples from the five points in each quadrat were mixed evenly and the indices of soil organic matter (SOM), mechanical composition (MC), total nitrogen (TN), total phosphorus (TP), total potassium (TK), alkaline hydrolysis nitrogen (AHN), available phosphorus (AP), available potassium (AK), and pH were determined, according to the industry standard 'LY/T1275-1999' [44].

*2.5. Statistical Analyses*

The Kruskal–Wallis non-parametric analysis of variance was used to test whether there were any significant differences in plant species diversity or soil factors between the

different rocky desertification degrees and then an LSD post-test was used. The correlations between species diversity and soil factors were analyzed using the Pearson correlation analysis. The above analyses and visualizations were performed on SPSS 25.0 (SPSS Inc., Chicago, IL, USA) and OriginPro 2022 (OriginLab, Hampton, MA, USA). The CCA analysis of the plant distribution and environmental factors was performed in R version 4.1.2.

## 3. Results

### 3.1. Vegetation Structure and Diversity

#### 3.1.1. Species Composition and Importance Values

The vegetation surveys discovered 247 species of woody plants in the study areas, which were from 149 genera and 62 families. The surveys also revealed the trend of plant diversity gradually increased as rocky desertification progressed (Figure 1). According to Table 2, the majority of the species belonged to the Pinaceae, Lauraceae, Fagaceae, Oleaceae, and Ulmaceae families, indicating that they were better suited to the unique rocky desertification conditions. Among them, several species of Oleaceae, Lauraceae, and Ulmus were distributed in different rocky desertification areas, whereas Cinnamomum and Machilus had diverse distributions, demonstrating that the same family of plants could adapt differently to rocky desertification environments. *Cinnamomum burmannii* and *Cinnamomum camphora* were distributed differently in various rocky desertification areas, indicating that the adaptability of the same genus to different rocky desertification environments was also varied. *Cinnamomum camphora* and *Schima superba* were only distributed in potential rocky desertification areas, *Pinus massoniana* was only distributed in potential, mild, and moderate rocky desertification areas, while *Acer coriaceifolium* was only distributed in severe and extremely severe rocky desertification areas, and *Quercus stewardiana* was only distributed in extremely severe rocky desertification areas. This shows that *Cinnamomum camphora*, *Schima superba*, *Pinus massoniana*, *Quercus stewardiana*, and *Acer coriaceifolium* were sensitive to changes in rocky desertification degree. So, they could be used as indicator plants for the succession process of rocky desertification.

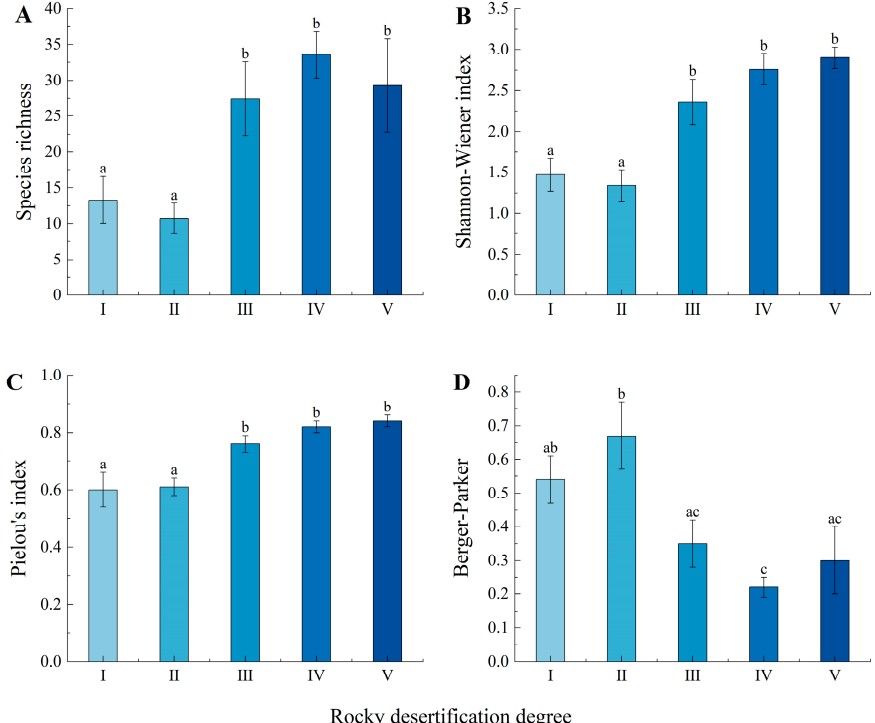

**Figure 1.** The species diversity indices of plant communities in plots with different rocky desertification degrees in northern Guangdong. Note: (**A**) species richness; (**B**) Shannon–Wiener index; (**C**) Pielou's index; (**D**) Berger–Parker index. Sites without the same letter are significantly different.

**Table 2.** The species and importance values of woody plants in areas with different rocky desertification degrees.

| Family Name | Species Name | Importance Value (%) | | | | |
|---|---|---|---|---|---|---|
| | | I | II | III | IV | V |
| Pinaceae | *Pinus massoniana* | 7.57 | 35.33 | 4.43 | - | - |
| Lauraceae | *Cinnamomum burmannii* | 15.82 | 18.27 | - | 0.17 | - |
| Fagaceae | *Castanopsis sclerophylla* | 7.50 | - | 20.10 | - | - |
| Oleaceae | *Osmanthus fragrans* | 5.99 | 2.96 | 1.59 | 2.75 | 2.86 |
| Lauraceae | *Machilus chinensis* | 2.42 | 0.75 | 5.35 | 2.21 | 3.29 |
| Ulmaceae | *Celtis sinensis* | 1.44 | 0.86 | 0.61 | 5.99 | 3.28 |
| Euphorbiaceae | *Mallotus philippensis* | 0.74 | - | 4.00 | 2.53 | 4.05 |
| Lauraceae | *Cinnamomum camphora* | 10.77 | - | - | - | - |
| Cornaceae | *Cornus wilsoniana* | 1.01 | - | 4.75 | 8.73 | 0.56 |
| Fagaceae | *Quercus stewardiana* | - | - | - | - | 7.49 |
| Sapindaceae | *Dimocarpus longan* | 0.38 | 4.70 | - | 0.81 | - |
| Aceraceae | *Acer coriaceifolium* | - | - | - | 0.50 | 5.31 |
| Verbenaceae | *Vitex negundo* | - | 1.90 | 0.64 | 1.72 | 1.25 |
| Leguminosae | *Zenia insignis* | - | 4.54 | - | - | 0.86 |
| Rhamnaceae | *Hovenia acerba* | - | 1.54 | - | 2.65 | 1.01 |
| Fagaceae | *Quercus glauca* | - | - | 0.20 | 2.84 | 1.98 |
| Hamamelidaceae | *Liquidambar formosana* | - | 1.37 | 1.15 | 1.41 | 1.01 |
| Theaceae | *Schima superba* | 4.58 | - | - | - | - |
| Anacardiaceae | *Choerospondias axillaris* | 1.56 | 0.79 | 0.29 | 0.76 | 2.47 |
| Hamamelidaceae | *Loropetalum chinense* | 1.52 | - | 1.26 | 0.25 | 1.19 |
| Anacardiaceae | *Pistacia chinensis* | - | - | 1.45 | 2.85 | 0.90 |
| Sterculiaceae | *Sterculia lanceolata* | - | 3.03 | - | 0.89 | - |
| Pittosporaceae | *Pittosporum illicioides* | - | - | 1.90 | 1.65 | 0.21 |
| Bignoniaceae | *Catalpa fargesii* | - | - | - | - | 3.72 |
| Araliaceae | *Kalopanax septemlobus* | 0.74 | 1.22 | - | 0.72 | 0.84 |
| Aquifoliaceae | *Ilex chinensis* | 0.64 | - | 0.71 | 1.56 | 0.59 |
| Fagaceae | *Quercus fabri* | - | - | 3.37 | - | - |
| Fagaceae | *Castanopsis faberi* | - | - | - | 3.26 | - |
| Cornaceae | *Alangium chinense* | - | 0.76 | - | 0.32 | 2.18 |

Note: Only species with a combined importance value of >3% are shown in the table; '-' means no such species in this plot.

The importance values of the plant species in the different RDD areas were studied (Table 2). A total of 29 species, or 11.3% of all species, had importance values greater than 3%. Different RDDs resulted in different dominant main species in the plant communities: *Cinnamomum burmannii*, *Cinnamomum camphora*, *Pinus massoniana,* and *Castanopsis sclerophylla* were the dominant species in potential rocky desertification areas; *Pinus massoniana* and *Cinnamomum burmannii* were dominant in mild rocky desertification areas; *Castanopsis sclerophylla* and *Machilus chinensis* were dominant in moderate rocky desertification areas; *Cornus wilsoniana* and *Celtis sinensis* were dominant in severe rocky desertification areas; and *Quercus stewardiana* and *Acer coriaceifolium* were dominant in extremely severe rocky desertification areas. Our analysis of endemic plants in areas with different RDDs showed that *Schima superba* was only distributed in potential rocky desertification areas, *Quercus fabri* was only distributed in moderate rocky desertification areas, *Castanopsis faberi* was only distributed in severe rocky desertification areas, and *Catalpa fargesii* was only distributed in extremely severe rocky desertification areas. Additionally, this analysis revealed that the importance values of the above species were low, indicating that these species were at a disadvantage in rocky desertification environments.

### 3.1.2. Plant Diversity

There were significant differences in species richness between the areas with different RDDs (Figure 1A); for example, the difference between the potential rocky desertification area and the mild rocky desertification area was not significant, which was due to the

progressive extinction of certain species, such as *Cinnamomum camphora* and *Schima superba*, that could not adapt to the severe rocky desertification environment, and the emergence of plants that could adapt, such as *Zenia insignis* and *Sterculia lanceolata*. The advent of plants suited for rocky desertification environments, such as *Quercus glauca* and *Pistacia chinensis*, could explain why species richness was substantially greater in moderate rocky desertification areas than mild rocky desertification areas. There was no discernible increase in species richness in severe rocky desertification areas compared to moderate rocky desertification areas, which was caused by a decrease in plants that could not adapt to severe rocky desertification environments, such as *Pinus massoniana* and *Castanopsis sclerophylla*, and the increase in plants that could adapt, such as *Dimocarpus longan* and *Acer coriaceifolium*. Species richness in the extremely severe rocky desertification areas did not differ significantly from that in severe rocky desertification areas, which could be related to the decrease in plants that could not survive in extremely severe rocky desertification environments, such as *Cinnamomum burmannii* and *Dimocarpus longan*, and the emergence of plants that could survive, such as *Quercus stewardiana*. The Shannon–Wiener diversity indices (Figure 1B) and Pielou's indices (Figure 1C) of the areas with different RDDs differed significantly and displayed upward trends. Additionally, these indices were significantly lower in potential rocky desertification and mild rocky desertification areas than moderate, severe, and extremely severe rocky desertification areas. The Berger–Parker indices of the areas with different RDDs were also significantly different (Figure 1D), with an overall downward trend, contrary to the trend of species richness. The Berger–Parker index was significantly lower in severe rocky desertification areas than potential rocky desertification areas and the Berger–Parker indices of moderate, severe, and extremely severe rocky desertification areas were significantly lower than that of mild rocky desertification areas. This shows that there were fewer dominant species in mild rocky desertification areas, the dominant species was the most obvious species, and the distribution was uneven. Meanwhile, in severe rocky desertification areas, there were numerous dominant species, their relative dominance was not clear, and their distributions were uniform.

### 3.2. Soil Factors

#### 3.2.1. Soil Physical Factors

There were no significant differences in soil moisture content between the areas with different RDDs in northern Guangdong; however, there were significant differences in bulk density and soil mechanical composition (Table 3). The soil moisture contents of the different rocky desertification plots were in the range of 16.90–20.78 g/cm$^3$, and there is no significant difference with the increase in rocky desertification. The bulk densities of the areas with different RDDs were in the range of 1.34–1.44 g/cm$^3$. With the deepening of rocky desertification, there is little difference in bulk density, but mildly rocky areas had higher bulk density than extremely severe rocky areas. In terms of soil mechanical composition, the number of 2–0.05 mm sand particles fluctuated with the increase in rocky desertification, with the highest contents observed in severe rocky desertification areas and the lowest observed in moderate rocky desertification areas. The number of 0.05–0.002 mm silt particles gradually decrease as rocky desertification progressed. Potential rocky desertification areas had the highest contents of these particles, while severe rocky desertification areas had the lowest contents. The number of <0.002 mm clay particles fluctuated with the increase in rocky desertification, with the highest contents observed in moderate rocky desertification areas and the lowest contents observed in severe rocky desertification areas.

#### 3.2.2. Soil Chemical Factors

There were no significant differences in soil pH (5.71–6.30), TP (0.29–0.45), TK (13.34–15.84), or AP (0.37–1.51) between the areas with different RDDs in northern Guangdong; however, there were significant differences in SOM (17.67–36.20), TN (1.30–2.49), AHN (60.29–148.87), and AK (29.23–48.54). The SOM, TN, AHN, and AK values increased in totality with the increase in rocky desertification (Table 4). The SOM contents of mild rocky desertification

areas were significantly lower than those of severe and extremely severe rocky desertification areas, while the SOM contents of extremely severe rocky desertification areas were significantly higher than those of the other four RDDs. There were no significant differences between the other RDDs. The TN of potential, mild, and moderate rocky desertification areas was significantly lower than that of severe and extremely severe rocky desertification areas. The AHN of potential and mild rocky desertification areas was significantly lower than that of severe and extremely severe rocky desertification areas, but there were no significant differences between moderate rocky desertification areas and the other RDDs. The AK of potential, mild, and moderate rocky desertification areas was significantly lower than that of extremely severe rocky desertification areas, but there were no significant differences between severe rocky desertification areas and the other RDDs (Table 4). According to the classification standard of the second national soil census, the SOM and TN of potential rocky desertification areas belonged to the third national nutrient level, while the TK and AHN belonged to the fourth level, TP belonged to the fifth level, and AP and AK belonged to the sixth level. The SOM, TN, and AHN levels all showed the tendency to increase totality, and the TK levels did not change significantly. However, the TP, AP, and AK levels were low in all five RDDs and were in the fifth and sixth nutrient levels.

**Table 3.** A comparison of the soil physical properties in areas with different rocky desertification degrees in northern Guangdong.

| RDD | Soil Moisture Content (g/cm$^3$) | Soil Bulk Density (g/cm$^3$) | Soil Mechanical Composition (%) | | |
| --- | --- | --- | --- | --- | --- |
| | | | 2–0.05 mm | 0.05–0.002 mm | <0.002 mm |
| I | 19.94 ± 1.35 a | 1.38 ± 0.04 ab | 35.00 ± 1.66 ab | 46.14 ± 1.42 a | 18.86 ± 2.32 a |
| II | 16.98 ± 1.59 a | 1.44 ± 0.04 a | 42.43 ± 2.48 bc | 40.14 ± 1.26 b | 17.43 ± 3.31 ab |
| III | 20.78 ± 0.98 a | 1.35 ± 0.03 ab | 32.67 ± 6.05 a | 38.67 ± 3.23 b | 28.67 ± 4.51 c |
| IV | 16.90 ± 3.11 a | 1.35 ± 0.04 ab | 53.80 ± 1.25 cd | 36.41 ± 1.20 b | 9.79 ± 0.52 b |
| V | 17.68 ± 1.06 a | 1.34 ± 0.02 b | 47.86 ± 3.36 d | 37.71 ± 2.13 b | 14.43 ± 1.81 ab |

Note: Sites without the same letter are significantly different.

**Table 4.** A comparison of the soil chemical properties in areas with different rocky desertification degrees in northern Guangdong. Note: Rocky desertification degree (RDD), pH value (pH), Soil organic matter (SOM), Total nitrogen (TN), Total phosphorus (TP), Total kalium (TK), Alkaline hydrolysis nitrogen (AHN), Available phosphorus (AP), Available kalium (AK). The same below.

| RDD | pH | SOM (g/kg) | TN (g/kg) | TP (g/kg) | TK (g/kg) | AHN (mg/kg) | AP (mg/kg) | AK (mg/kg) |
| --- | --- | --- | --- | --- | --- | --- | --- | --- |
| I | 6.30 ± 0.29 a | 21.35 ± 1.91 ab | 1.38 ± 0.12 a | 0.30 ± 0.06 a | 14.60 ± 2.62 a | 75.45 ± 7.02 a | 0.41 ± 0.09 a | 29.49 ± 4.18 a |
| II | 6.30 ± 0.27 a | 17.67 ± 1.80 a | 1.30 ± 0.13 a | 0.29 ± 0.06 a | 15.84 ± 2.07 a | 60.29 ± 3.72 a | 1.11 ± 0.80 a | 34.41 ± 1.38 a |
| III | 6.20 ± 0.18 a | 21.67 ± 3.40 ab | 1.39 ± 0.23 a | 0.31 ± 0.02 a | 13.34 ± 2.26 a | 83.77 ± 17.01 ab | 0.37 ± 0.05 a | 29.23 ± 2.41 a |
| IV | 6.09 ± 0.20 a | 28.50 ± 1.89 b | 1.97 ± 0.22 b | 0.38 ± 0.10 a | 15.10 ± 2.97 a | 111.14 ± 8.16 b | 1.51 ± 0.92 a | 41.53 ± 6.67 ab |
| V | 5.71 ± 0.12 a | 36.20 ± 3.05 c | 2.49 ± 0.26 b | 0.45 ± 0.05 a | 15.75 ± 1.35 a | 148.87 ± 10.40 c | 0.79 ± 0.16 a | 48.54 ± 5.46 b |

Note: Sites without the same letter are significantly different.

### 3.3. Relationships between Soil Factors and Plant Community Structures

3.3.1. Correlation Analysis

In the areas with different RDDs, our correlation analysis of plant diversity and soil factors revealed that soil pH, SOM, TN, and AHN were significantly correlated with plant diversity, but SMC, SBD, TP, TK, AP, and AK were not. (Figure 2). Significantly positive correlations were found between RDD and SOM, TN, AHN, and AK levels. The Shannon–Wiener index had a statistically significant inverse relationship with pH, a significantly positive relationship with SOM and TN, and a highly significantly positive relationship with AHN. The Pielou's index was significantly positively correlated with TN and highly significantly positively correlated with SOM and AHN. The Berger–Parker index was significantly positively correlated with pH, significantly negatively correlated with TN, and highly significantly negatively correlated with SOM and AHN. Species richness was significantly negatively correlated with pH, significantly positively correlated with SOM,

and highly significantly positively correlated with AHN. As can be observed, there were considerable correlations between plant diversity and the pH, SOM, TN, and AHN levels of the karst rocky desertification soil in northern Guangdong.

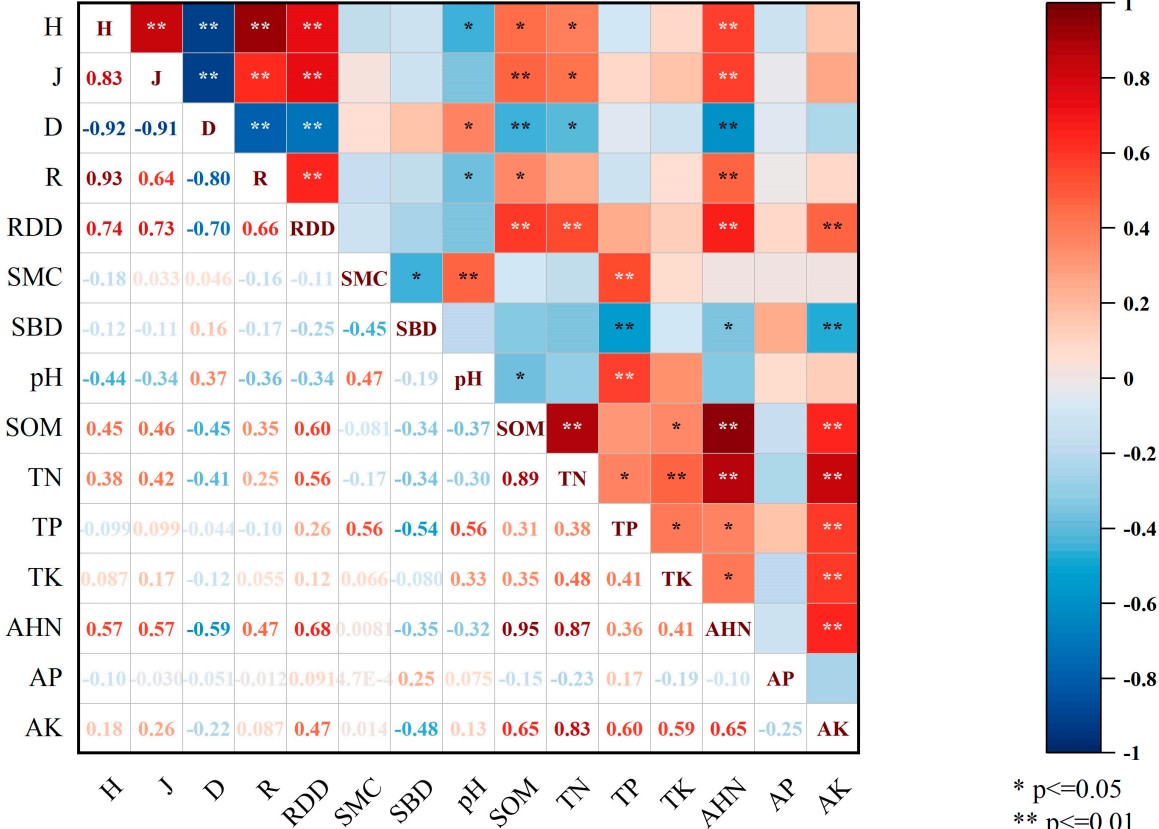

**Figure 2.** The correlations between the plant diversity index and soil physicochemical properties in the study plots with different rocky desertification degrees. Note: Shannon–Wiener index (H), Pielou's index (J), Berger–Parker index (D), species richness (R), Rocky desertification degree (RDD), Soil moisture content (SMC), Soil bulk density (SBD), Soil organic matter (SOM), Total nitrogen (TN), Total phosphorus (TP), Total kalium (TK), Alka-line hydrolysis nitrogen (AHN), Available phosphorus (AP), Available kalium (AK). *, $p \leq 0.05$; **, $p \leq 0.01$. The same below.

### 3.3.2. CCA Analysis of Plant Communities and Soil Factors

The CCA analysis of the species and environmental factors in the rocky desertification plots produced a two-dimensional ordination diagram (Figure 3), in which the arrows represent various environmental parameters, and the lengths of the arrows indicate the correlations between species distributions and environmental conditions. The first two ordination axes had a cumulative contribution rate of 42.5%, while the first five ordination axes had a cumulative contribution rate of 87.9%. The eigenvalues were 0.715 and 0.662, respectively (Table 5), which were significantly correlated with the environmental factors, indicating that the first two axes could better reflect the relationships between species distributions and environments. From Figure 3 and Table 5, it can be seen that the factors with the greatest correlation with the first axis were rocky desertification degree and TP, with correlation coefficients of 0.495 and −0.435, respectively. AP had a correlation coefficient of −0.492, making it the factor that was most correlated with the second axis. These findings indicated that TP, AP, and rocky desertification degree were the primary environmental factors that determined the species distributions. The species coordinates close to the center of the plot showed that the species was highly adaptable within the whole plot. On the contrary, this also suggested that the species could only thrive in a particular setting. Coordinates that were similar implied similar growing environments. As can be seen in Figure 3, the

species *Ilex aculeolata* and *Dalbergia benthamii* were located at the edge of the first quadrat as they are suitable for significantly sloping environments, whereas *Glycosmis pentaphylla*, *Tinospora sagittata*, *Endospermum chinense*, *Melodinus cochinchinensis*, *Sterculia lanceolata*, and *Adenanthera microsperma* were located at the edge of the third quadrat as they are suitable for high TP soil environments.

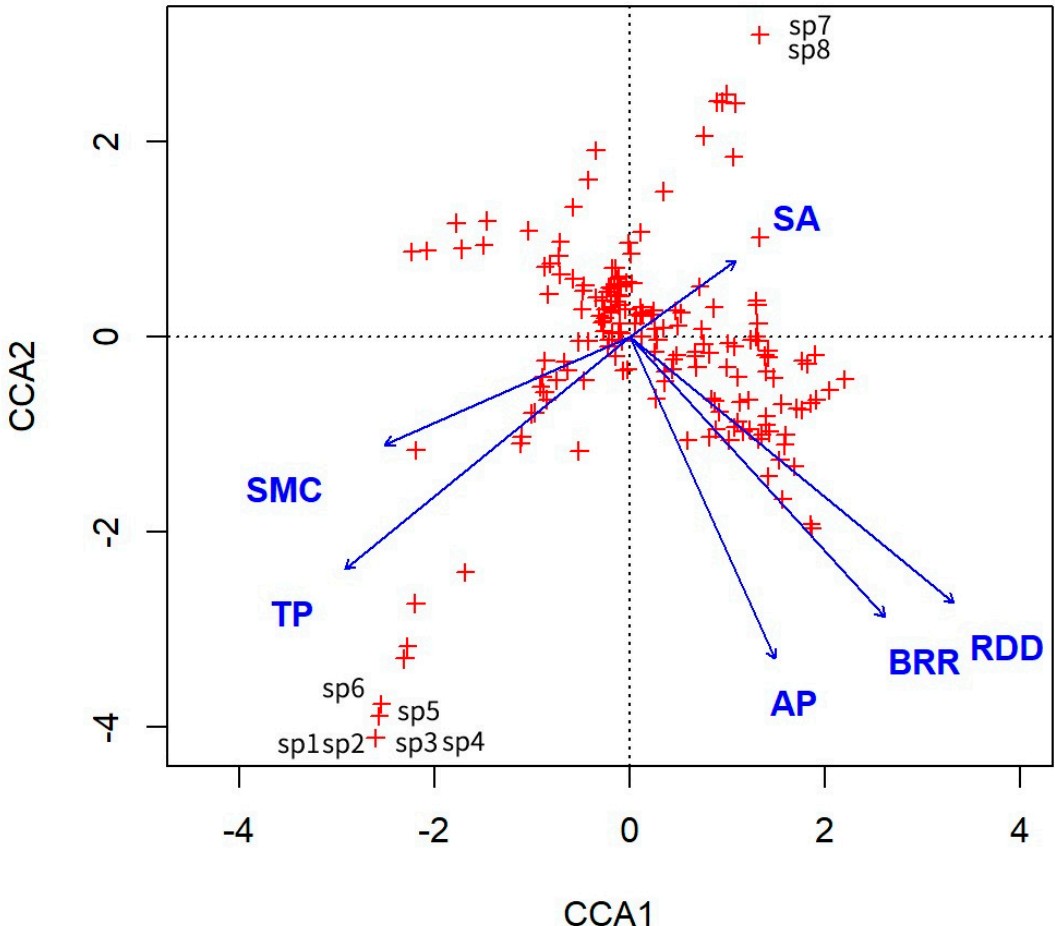

**Figure 3.** The CCA ordination diagram of plant communities and environments in the rocky desertification study plots in northern Guangdong. The blue arrows represent various environmental parameters. The red "+" represents the distribution of different species Note: slope (SA), bedrock exposed rate (BRR). sp1, *Glycosmis pentaphylla*; sp2, *Tinospora sagittate*; sp3, *Endospermum chinense*; sp4, *Melodinus cochinchinensis*; sp5, *Sterculia lanceolata*; sp6, *Adenanthera microsperma*; sp7, *Ilex aculeolate*; sp8, *Dalbergia benthamii*.

**Table 5.** The eigenvalues of the CCA ordination axes and their correlations with the environmental factors.

|  | **CCA1** | **CCA2** |
|---|---|---|
| Eigenvalue | 0.7145 | 0.6618 |
| Proportion Explained | 0.2207 | 0.2044 |
| Cumulative Proportion | 0.2207 | 0.4251 |
| RDD | 0.4952 | −0.4068 |
| TP | −0.4353 | −0.3551 |
| AP | 0.2219 | −0.4922 |
| SMC | −0.3747 | −0.1647 |
| BRR | 0.3893 | −0.4277 |
| SA | 0.1622 | 0.1164 |

## 4. Discussion

### 4.1. Response of Plant Community Structure Characteristics to Different RDDs

Plant species diversity has always been a focus for forest ecologists and the basis of species research [45]. The vegetation of ecosystems in the rocky desertification areas in northern Guangdong is generally rich. We identified 247 species of woody plants, spanning 62 families and 149 genera. However, in extremely severe rocky desertification areas, there were only 85 species of woody plants, spanning 42 families and 65 genera. The species richness was lower in extremely severe rocky desertification areas than in other RDDs, indicating that plants responded strongly to the rocky desertification environments. According to earlier studies [46,47], this could be caused by the erosion of rocky arid soil and the significant amount of exposed bedrock [39]. Several Oleaceae, Lauraceae, and Ulmaceae genera and species were found in the sample plots and their importance values were high, indicating that they were better able to adapt to these environments than other plant groups, although plants belonging to different genera within the same family demonstrated varying degrees of rocky desertification adaptations. It can be seen from the importance values that with the increase in rocky desertification, the plants that were not suitable for high RDD environments gradually disappeared, including *Cinnamomum camphora*, *Schima superba*, *Pinus massoniana,* and *Castanopsis sclerophylla*, while plants that were suitable for high RDD environments gradually increased, such as *Quercus glauca*, *Acer coriaceifolium*, *Quercus stewardiana*, etc. This demonstrated that the plants that were suitable for rocky desertification areas had stony, drought-tolerant, and calcium-preferential features, as well as strong adaptability to environmental factors [48]. In conclusion, the rocky desertification areas in northern Guangdong possessed different plant community structure characteristics than those of subtropical conventional plant communities [49], as well as other rocky desertification areas [50].

The changes in species richness, diversity index, and evenness index revealed that the diversity index and the species richness showed the trend of gradually increasing with the progression of rock desertification, indicating that the rock desertification management in northern Guangdong has achieved certain results. Meanwhile, the plant community structures first degraded and then gradually improved with the succession of rocky desertification, and then the community structures tended to become stable; the Berger–Parker indices verified this result. The trends in soil chemical characteristics could be connected to the changes in diversity index. Soil nutrients can promote vegetation growth at certain concentrations [20], but vegetation has a certain tolerance to soil, so soil nutrient concentrations that are too high or too low can instead inhibit vegetation growth and result in a decrease in vegetation cover, which is why potential rocky desertification environments had lower plant diversity indices [51]. In conclusion, the rocky desertification degree was significantly correlated with species richness, diversity index, evenness index, and Berger–Parker index; however, according to our CCA analysis of vegetation and environmental factors, the rocky desertification degree had a greater influence on plant community structures.

### 4.2. Response of Soil Factors to Different RDDs

Numerous factors affect soil, including environmental elements (such as climate, parent material, and terrain) and anthropogenic influences (such as tillage and grazing) [52]. Rocky desertification soil develops from carbonate rocks, which are rich in calcium and magnesium and can obtain higher accumulations of SOM and TN [53,54]. The forest vegetation in potential rocky desertification areas is subject to broken rings [55], which inhibit plants from fixing soil and reduce litterfall, thereby resulting in the prevention of SOM accumulation in the soil [56], as well as the loss of other nutrients, such as TN and AHN, through soil erosion. This degrades the soil [57]. The effects of erosion weaken with the progression of rock desertification [34], corresponding to decreases in soil nutrient losses, more apparent bedrock aggregation effects, and the accumulation and storage of organic matter and nitrogen in the soil, which improve soil nutrient levels over time [58]. According to our findings, there were significant differences in SBD, MC, SOM, TN, AHN, and AK

levels between the different RDDs, indicating that these soil factors were more influenced by the RDD. However, there were no significant differences in SMC, pH, TP, TK, or AP between the different RDDs. The varying sample collection times, which were significantly impacted by precipitation, could have been the cause of the changes in moisture content. In northern Guangdong, rainfall is plentiful and the rainy season lasts from May to September [59], so soil moisture content measurements that are taken during periods of heavy precipitation are generally high. In our study, as the RDD increased, the soil bulk density showed a small change trend. We also found that the soil bulk densities in mild rocky desertification areas were larger and the soil nutrient losses were more serious, which was consistent with the findings of previous studies [7], indicating that the soil nutrient contents in mild rock desertification areas were the lowest. The number of large-diameter soil particles fluctuates and increases with the deepening of rocky desertification. The number of large-diameter soil particles in severe rock desertification areas increased by 18.8% in comparison to that in potential rock desertification areas, which was brought on by soil erosion from the rock desertification process [60]. The soil chemical properties (except pH) did not always decrease as rocky desertification progressed, but rather showed a fluctuating trend of increase. Some of the soil nutrients in extremely severe rocky desertification areas were better than those in mild rocky desertification areas, demonstrating the contrary trend as SBD, such as SOM, TN, AHN, and AK. This was consistent with earlier studies [7]. This was due to the aggregation effect of rocky desertification on nutrients [58]. Among them, SOM, TN, and AHN levels increased significantly overall, demonstrating that SOM and N had a strong response to the RDD. Meanwhile, TP, AP, and AK contents were in the range of nutrient deficiency in potential rocky desertification areas, which was primarily influenced by the soil-forming parent and did not improve during the rocky desertification process, indicating that P and K were limiting factors in the process of rocky desertification soil restoration.

*4.3. Relationships between Plant Community Structures and Soil Factors*

Ecologists have long focused on the connections between plant communities and soil environments, and some research has discovered substantial associations between plant community structural characteristics and soil nutrients [61,62]. The findings of this study also demonstrated substantial relationships between species diversity and soil factors in different rocky desertification areas. The Shannon–Wiener diversity index showed a significantly negative correlation with pH, a significantly positive correlation with SOM and TN, and a highly significantly positive correlation with AHN. The Pielou's index showed a significantly positive correlation with TN and a highly significantly positive correlation with SOM and AHN. The Berger–Parker index showed a significantly positive correlation with pH, a significantly negative correlation with TN, and a highly significantly negative correlation with SOM and AHN. Finally, species richness was significantly negatively correlated with pH, significantly positively correlated with SOM, and highly significantly positively correlated with AHN. This indicated that some soil nutrients and plant community structures were enhanced during the succession of rocky desertification, which was consistent with previous studies [11,63].

Plant communities are the consequences of long-term interactions and adaptations between species and environmental factors [64]. According to our CCA findings, TP and AP were the key soil variables that influenced how plants were distributed in different rocky desertification areas. Additionally, soil phosphorus content was influenced by a combination of soil formation processes, soil properties, and external disturbances [17]. Significant variations in soil nutrients existed between the regions with different RDDs, particularly between prospective or moderate rocky desertification areas and severe rocky desertification areas [7]. In contrast, there were no statistically significant differences in TP or AP levels between the areas with different RDDs, which was due to the poor soil in the study region. According to the second national soil survey classification standard, AP belongs to the sixth national soil nutrient level and TP belongs to the fifth level, which

are far below the national average [65]. Due to the low levels of AP in the potential rocky desertification areas in this study, nutrient enrichment during soil succession was prevented, making TP and AP the primary soil variables that influenced local plant distributions. Therefore, the TP and AP contents were significant markers for soil restoration in rocky desertification areas.

## 5. Conclusions

Our study of the plants and soil in our research areas revealed that there were differences in the adaptability of species to different RDDs. Oleaceae, Lauraceae, and Ulmaceae could be considered pioneer species since they were more tolerant of rock desertification environments. *Cinnamomum camphora*, *Schima superba*, *Pinus massoniana*, *Quercus stewardiana*, and *Acer camphora* had substantial responses to changes in RDD; therefore, they could be used as indicator species. The right species should be chosen for rocky desertification restoration based on the RDD.

There were also significant differences in plant community structural characteristics and soil factors between the different rocky desertification areas in northern Guangdong. The vegetation generally exhibited the trend of first degrading and then gradually improving and stabilizing. Additionally, the soil chemical properties had significant aggregation effects, in which the contents of SOM, TN, and AHN are gradually increasing, which was the same as the change trend of the vegetation, and there were significant correlations between them. This indicated that there was a strong coupling relationship between vegetation and soil in the rocky desertification areas in northern Guangdong. The TP, AP, and AK contents were in the nutritionally deficient range in all of the rocky desertification areas. Therefore, supplementary P and K should be increased in rocky desertification restoration strategies. Our CCA analysis showed that the plant distributions in rocky desertification areas were greatly affected by TP and AP levels, and that increasing phosphorus fertilizer input could be a primary measure for land treatment. In summary, this study examined the changes in plant diversity and soil during the process of rocky desertification and analyzes the relationship between them. The results could be of great significance for revegetation and soil remediation strategies for the control of rocky desertification in northern Guangdong. Plants and soil are mutually affected, but their changes are also affected by other environmental factors. Therefore, future research needs to consider terrain, climate, and other factors to provide reference for the restoration and reconstruction of rocky desertification.

**Author Contributions:** M.L., C.X. and Z.S. conceived and designed the research. J.Y., N.W., C.S., G.W. and H.C. collected the data. M.L. analyzed the data and wrote the paper. Z.S. revised the paper. All authors have read and agreed to the published version of the manuscript.

**Funding:** This research was supported by the Forestry Department of Guangdong Province, China, for Non-Commercial Ecological Forest Studies (YLC.2021-1).

**Data Availability Statement:** Not applicable.

**Conflicts of Interest:** The authors declare that the research was conducted in the absence of any commercial or financial relationships that could be construed as potential conflict of interest.

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
