# Peer review of "Changes in Plant Diversity and Soil Factors under Different Rocky Desertification Degrees in Northern Guangdong, China"

_forests, doi:10.3390/f14040694_

Round 1

Reviewer 1 Report

Review Report for Manuscript Forests-2255709

The manuscript, entitled "Plant diversity and soil factors under different rocky desertification degrees in Northern Guangdong, China” is based on a novel idea but some issues. So, I recommend the "major revision" of this manuscript after handling the suggestions listed below.

  • Line 2-3. Section; Title. The title needs revision, and it should be based upon the aims of this study. Try to add role of this study in revegetation of rocky desertification.
  • Line 15. Section; Abstracts. “CCA analysis” kindly add full name of this analysis first and then later on in this paper write its abbreviation.
  • Line 29. Section; Abstracts. “Rocky deserts in northern Guangdong has yielded some interesting results” It is better to enlist these interesting results in abstract section for more clarity.
  • Line 35-36. Section; Key words. Arrange key words alphabetically for more clarity.
  • Line 50, 72. Section; Introduction. “Guangdong Province” should be revised as “Guangdong province”.
  • Line 87-89. Section; Introduction. The sentence “The soil–plant relation- 87 ship is an indivisible ----------- the former to some extent” should be revised for clarity. Use smaller sentences for better understanding.
  • Line 100-114. Section; Introduction. The aims and objectives of this study should be written clearly. Plant diversity and physiochemical properties of soil were studied in this research, but the link between these two factors and environmental factors were missing I think the rainfall pattern and altitude also plays a vital role in rocky desertification. Do you also think the studied factors have role in revegetation if yes kindly discuss it in this section.
  • Line 129-130. Section; Materials and methods. It is better to revise this “(20m × 20m)” as (20 × 20m)”
  • Line 129-130. Section; Results. “16.90-20.78g cm3” should be revised as 16.90-20.78 g cm3”.
  • Line 264. Section; Results. Revise this “0.002” statement as 0.002”.
  • Line 294-295. Section; Results. Add full names of the abbreviations used in table 4 at the end of table indent.
  • Line 168, 247, 296, 316, 349, 390, 429. Section; Results. Revise all headings of this section and use only capital letter for first word only while all other should be small.
  • Line 294-295. Section; Results. Revise these lines for clarity.
  • Line 313-315. Section; Discussion. Give details of all the abbreviations used in figure 2 at the end of figure indent.
  • Line 460. Section; Conclusion. Add future recommendations at the end of conclusion.
  • Line 494. Section; References. Cross-check all the enumerated references with the references cited in the text.

Reviewer 2 Report

The study described the role of different plant species in rocky desertification. The introduction part is well-written. The results are good; please correct the grammar error; plant names should be in italics. The discussion part is easy to follow and well-written. overall the study is informative.

Reviewer 3 Report

Abstract is good.

Introduction: Good review of relevant literature and clear statement of objectives.

Materials and Methods:
line 129,136: 20 m x 20 m
line 143 - what is relative advantage?
line 148, 156 - references to industry standards are not clear
line 150 ff- combine 2 sentences to read: The samples were put into sealed bags, taken back to the laboratory, and dried in an oven ...
Table 1. I would suggest using Average Slope rather than Average Gradient

Results:
Line 180 and following all one sentence. Divide into 3 sentences with first two sentences ending where "and" is currently used.

Some reference to Table 2 should be made in first paragraph of Results.

Figure 1 title: Just say, Sites without the same letter are significantly different.

Line 216 and following.  the statement, "it was lower in mid rocky desertification areas than potential rocky areas" is not a true statement based on statistics.

Line 235 and following: the Berger-Parker index does vary with RDDs, but does not initially increase and then decrease and increasing.

Line 252 - According to statistics, there is no difference in soil moisture (all a).
Line 253 - like wise bulk density does not increase and then decrease. Very little difference in bulk density. Mildly rocky areas had higher bulk density than extremely severe rocky areas.
Line 290-293 - SOM did not increase and then decrease; TN increases; TK doesn't change; AHN increases.
Line 334 - should refer to Fig. 3 not Fig. 4
Line 409 - soil bulk density did not increase and then decrease.
Line 413 - The number of larger diameter soil particles do not increase and decline.
Line 417-419 - Soil chemical properties do not follow the trend of first decreasing and then gradually increasing. If anything, they tend to increase.
Line 422 - TK does not change
Line 439 - The Berger_Parker index showed a highly significant negative correlation with DOM and AHN.

Conclusions:

Line 471-473 - The statement that SOM, TN, TK, and AHN showed the trend of first degrading and then gradually improving is not true based on statistics given.

References:

All citations in manuscript are listed in the References.

Line 551 - separate the word Forests and under

Round 2

Reviewer 3 Report

Most responses to previous comments are addressed, but a few concerns still exist.

Point 1. Are spacings when referring to size of plots, etc. just typos?
line 150-151, 157,173: 20 x 20 m
line 174: 0-30 cm

Point 11. lines 274-277. Reworded text still does not agree with statistics. There is no difference in soil moisture contents with the increase in rocky desertification.

Other comments:

line 127: Plant should not be capitalized

line 194-196 in reference to Fig. 1 - What measure of plant diversity initially decreases, then increases, and decreases again? The Shannon-Wiener and Pielou's index both indicate that increasing diversity after potential and mild levels of rocky desertification.

line 279-280: reword to say "With the deepening of rocky desertification, there is little difference in bulk density, but mildly rocky areas had higher bulk density than extremely rocky areas.

delete the next sentence (line 280-282) as it is redundant and not completely correct.

line 297: remove TP from list as it did not increase.

line 444-446: Either say some of the soil nutrients or specify which ones were better.  Do they demonstrate the same trend as SBD? Soil bulk density was less on severe rocky areas compared to mild rocky areas.

line 448-449: the phrase, "the soil nutrient initially degraded and then improved" does not seem correct based on statistics given in Table 4.
